# Collagen Structured Hydration

**DOI:** 10.3390/biom13121744

**Published:** 2023-12-04

**Authors:** Satyaranjan Biswal, Noam Agmon

**Affiliations:** The Fritz Haber Research Center, Institute of Chemistry, The Hebrew University of Jerusalem, Jerusalem 9190401, Israel; satyaranjan.biswal@mail.huji.ac.il

**Keywords:** collagen, water, hydration, radius of gyration, SASA, hydrogen bond, residence time

## Abstract

Collagen is a triple-helical protein unique to the extracellular matrix, conferring rigidity and stability to tissues such as bone and tendon. For the [(PPG)${}_{10}$]${}_{3}$ collagen-mimetic peptide at room temperature, our molecular dynamics simulations show that these properties result in a remarkably ordered first hydration layer of water molecules hydrogen bonded to the backbone carbonyl (bb-CO) oxygen atoms. This originates from the following observations. The radius of gyration attests that the PPG triplets are organized along a straight line, so that all triplets (excepting the ends) are equivalent. The solvent-accessible surface area (SASA) for the bb-CO oxygens shows a repetitive regularity for every triplet. This leads to water occupancy of the bb-CO sites following a similar regularity. In the crystal-phase X-ray data, as well as in our 100 K simulations, we observe a 0-2-1 water occupancy in the P-P-G triplet. Surprisingly, a similar (0-1.7-1) regularity is maintained in the liquid phase, in spite of the sub-nsec water exchange rates, because the bb-CO sites rarely remain vacant. The manifested ordered first-shell water molecules are expected to produce a cylindrical electrostatic potential around the peptide, to be investigated in future work.

## 1. Introduction

As far as we currently know, water is essential for life, particularly in its liquid state that allows dynamics to take place [1]. Indeed, hydrogen bonds (HBs) in liquid water form and break on a single-ps timescale. These HB fluctuations drive important processes such as ultrafast proton mobility in liquid water [2]. Consequently, “the presence of highly structured interfacial water correlated to specific functional states of proteins…surprised the community” [3]. As an example, Ref. [3] discussed X-ray diffraction from the snow flea anti-freeze protein (sfAFP). Although crystallized at *room temperature*, it exhibited “an array of highly ordered water molecules on the flat, hydrophobic face of the protein” [4]. While the family of antifreeze proteins, essential for life at subzero temperatures, is diverse, sfAFP has no known homologues. Its structure is based on a GXY repeat unit, where G = Gly (glycine), and X is often another Gly. Occasionally, X = Cys (cystein), leading to intramolecular SS bonds.

Interestingly, collagens also have GXY as their repeat units, though here X and Y are often the imino acids P = Pro (proline) or O = Hyp (hydroxyproline), respectively, so that Pro-Hyp-Gly (i.e., POG) is the most common triplet in collagen [5,6]. While this ends the formal similarity between the two proteins, we will show below, using molecular dynamics (MD) simulations, that a collagen-mimetic peptide (CMP) based on the PPG repeat unit also possesses highly ordered interfacial water molecules (WMs) at *ambient temperature*, with conjectured functional consequences.

Collagen is the most abundant structural protein in vertebrates, dominant in the extracellular matrix (ECM) and estimated to account for around 30 percent of the total protein weight in the human body. There are 28 known collagen families, among which collagens I, II, III, V, and XI are dominant and found in bones, tendons, cartilage, skin, blood vessels, and more [5]. As mentioned above, each collagen polypeptide chain has the ${\left(\mathrm{GXY}\right)}_{n}$ (or, equivalently, ${\left(\mathrm{XYG}\right)}_{n}$) structure, folding into a left-handed helix. Three staggered polypeptide chains then combine to form a right-handed triple helix, ${\left[{\left(\mathrm{GXY}\right)}_{n}\right]}_{3}$, with 7/2 or 10/3 helical symmetry (3.5 or 3.33 residues per turn, respectively) [7,8,9]. The three chains are linked together by strong HBs between the glycine’s NH group and the CO group at the X position of an adjacent chain, G-NH⋯OC-X. There is only this one direct HB between adjacent chains per triplet. When X and Y are imino acids then, due to their proline ring, there is no H atom bonded to their backbone (bb) N atom. When X and Y are amino acids, their NH group points outward from the triple helix axis. However, a X-NH⋯OH${}_{2}\cdots$OC-G single-water bridge can then form [10]. Weaker CH⋯OC HBs, emanating from an α-carbon, e.g., G-CH${}_{\alpha }\cdots$OC-X, also contribute to the triple helix stability [11].

Biological collagen samples, such as a tendon, consist of a hierarchy of collagen molecules in which triple helices assemble into microfibrils, which assemble into fibrils and finally cross-linked fibers [5]. These are difficult to disentangle and crystallize, so only low-resolution X-ray structures are available for natural collagen samples [12]. To circumvent this complexity, various CMPs have been synthesized and crystallized. The simplest are the (PPG)${}_{n}$ peptides (n = 9 or 10), whose X-ray structures have been determined several times [13,14,15,16,17,18,19] (see the Protein Data Bank (PDB) structures 1K6F, 1A3I, 1A3J, 2CUO, 1ITT, and 3AH9). These have higher resolutions than the earlier biological collagen samples, revealing not only the details of the triple helix, but also individual WMs at the peptide–water interface.

Unlike an earlier NMR study suggesting that an ordered “water wire” exists parallel to the collagen main axis [20], other ordered arrangements of water were revealed by peptide diffraction studies [21,22]:(a)The bb-carbonyl oxygen atoms of the G, X, and Y residues accepts HBs from 1, 0, and 2 WMs, respectively. Clearly, X-CO is not hydrated because it is the HB acceptor in a direct G-NH⋯OC-X HB.(b)In (PPG)${}_{n}$ X-ray structures [14,15], bb-carbonyls from Gly and Pro(Y) are connected by water bridges of one to three WMs.(c)When an amino acid is present in the X or Y positions, or Y is Hyp, their NH or OH moieties (respectively) serve as anchor points for additional water bridges.

Water bridges have fascinated the collagen X-ray community because they were envisioned as stabilizing the collagen structure [6,22,23,24]. These chains were “immobilized relative to bulk water and aligned in the plane transverse to the axis of the molecule”, thus explaining the “tendon magic angle effect” seen in magnetic resonance imaging [25]. This “ice-like” behavior must persist at least into the msec time regime to be observed in T2 NMR measurements [26]. Only one work has casted doubts on the water bridge explanation for collagen stability [27].

The question is to what extent are structures observed in crystals at cryogenic temperatures (e.g., 100 K) relevant to room-temperature liquid water? Clearly, water hexagons and cubes are seen in ices, which does not mean they characterize liquid water. Moreover, in liquid water the protein is surrounded by many more WMs as compared to its X-ray structure, and these could create networks that branch in all directions, obscuring any putative linear water chain. Such questions can be readily addressed by MD simulations, a field benefiting from constant improvement in force fields and analysis tools.

Indeed, several MD simulations of collagen-like peptides were conducted in search of stable water bridges [28,29,30,31,32,33,34]. For concreteness, consider the last study on the list, by Madhavi et al. [34], which is the most recent and detailed. It considered a triple helix composed of 30-residue peptide chains (PDB file 1BKV [35]), in which the first three and last four triplets are (POG)${}_{3}$ and (POG)${}_{4}$, respectively. The central portion of this peptide contains (non-imino) amino acids, and is less relevant for the present study. They analyzed all single-water bridges, which in the imino-rich segments can be only of the intramolecular G-CO⋯HOH⋯OC-Y type. With a cutoff of 2.4 Å for the OH⋯O HB distance, this bridge was found in only 1.2% of the 10 ns MD frames at 310 K (quite in contrast to an earlier study [32]). Thus, longer bridges (not monitored) may be even less probable. We conclude that water bridges are insignificant for the (PPG)${}_{10}$ peptide at room temperature.

In contrast, first-shell hydration (involving the bb-carbonyl oxygen atoms) is significant. As mentioned above, the X-ray studies indicated 0-2-1 hydration for the X-Y-G triplet at 100 K. Adding the single-water and (twice the) double-water hydration probabilities at 310 K from Tbl. 6 of Madhavi et al. [34] gives hydration numbers of <0.01, 1.26, and 0.99, respectively. While this qualitatively resembles the X-ray results, the MD study reported only the average over all imino-rich sites. Moreover, the (continuous) residence times of water hydrogen atoms at the Y-CO and G-CO sites were found to be only 0.84 and 4 ps, respectively. These numbers are similar to (or shorter than) those of bulk liquid water, thus giving no hint of an “ice-like” behavior.

Nevertheless, there are several arguments suggesting that these residence times should be longer:(a)The PPG segments are short and located at the two ends of the 1BKV peptide, where the triple helix is less compact and more exposed to solution.(b)The zwitterionic peptide was not “capped” to better mimic infinitely long chains.(c)The 10 ns simulations are not sufficiently long to obtain the residence time distribution (only its average).(d)The average residence time was measured for a water H atom, rather than its O atom. This monitors water reorientation times, which terminate with a flip exchanging the two H atoms of a WM rather than the O atom leaving its binding location. Consequently, the study is best repeated for an intact (PPG)${}_{n}$ peptide.

The present work analyses classical MD trajectories for the [(PPG)${}_{10}$)]${}_{3}$ collagen-like peptide, determining the structure of its hydration layers. We show that at 100 K an X-ray 0-2-1 hydration pattern is observed. At room temperature, in spite of the rapid water dynamics, the first hydration layer is nearly as ordered as in the crystal state, with periodic 0-1.7-1 hydration for all X-Y-G triplets. Similar periodic behavior is calculated for the solvent-accessible surface area (SASA). The average continuous residence times for a WM in the G-CO and Y-CO sites are about 82 and 26 ps, respectively, indicating that water exchange between the first hydration shell and the bulk is actually much slower than in pure water.

## 2. Materials and Methods

For collagen model peptides, X-ray crystallography has provided high-resolution structures of the basic triple-helical conformation and its water-mediated hydration network [15,18,19]. In our study, the crystal structure of the CMP [(PPG)${}_{10}$)]${}_{3}$, available in the Protein Data Bank (PDB ID: 1K6F), was used for the MD simulations [17]. This peptide was chemically synthesized, and its X-ray diffraction was taken at 1.30 Å resolution [17]. It was previously studied for properties such as stability and solvation of collagen protein [31,36,37]. The unit cell consisted of two head-to-tail triple helices, of which we focused on one (chains A, B, and C).

The two termini of the tripeptide are oppositely charged (the N-terminal positive and C-terminal negative). Consequently, they showed unusual interaction with solvent molecules during the simulation, including partial unwinding. Hence, we capped the two ends with N-terminal acetyl and C-terminal N-Me amide to neutralize their respective terminals by using Amber Tools [38], performing the MD under both the capped and uncapped conditions.

The choice of the force-field (FF) for the protein and its water solvent significantly impacts MD simulation results. Here, we utilize the AMBER14SB FF for the protein [39], in comparison with the CHARMM36m FF [40] (Appendix A). For the explicit water model, we use the TIP4P/2005, which reproduces accurately the water density, enthalpy of vaporization, and oxygen–oxygen radial distribution function at room temperature [41]. As the water model may be crucial for hydration properties, we compare the results with those from the TIP3P models (Appendix A).

All simulations were conducted using the GROMACS MD software, version 2020.2 [42], at three different temperatures: 300, 250, and 100 K. The 168 crystal WMs in the X-ray structure were either deleted or retained, which appeared to have little effect on the equilibrated structure of the hydration layers because the water ligands are readily exchanged with the bulk [43,44,45]. To run the simulation, we have considered different simulation boxes containing between 12,000 and 62,306 WMs with appropriate density. A triclinic box was used to define the systems’ boundaries because of the peptide’s cylindrical shape [46]. The protein was separated by at least 10 Å from the box boundaries in all directions. Periodic boundary conditions were imposed, with particle mesh Ewald (PME) summation to evaluate the long-range Coulombic interactions [47].

At 300 and 250 K, the subsequent equilibration phase comprised of three steps. First, energy minimization was performed, followed by 1 ns equilibration in the NVT ensemble (constant volume, temperature, and number of particles) and finally 1 ns equilibration in the NPT ensemble (constant pressure, temperature, and number of particles), using a Parrinello–Rahman barostat at 1 atm [48]. This recommended order (NVT followed by NPT equilibration) in the Gromacs tutorial (http://www.mdtutorials.com/gmx/lysozyme/07_equil2.html, accessed on 22 November 2023) supposedly results in fast thermal equilibration in the NVT step, which can then be followed by pressure equilibration in the NPT step. For biological systems, it is observed that with the absence of the NVT step, enormous pressures may develop at the onset of the NPT step, pushing around water molecules into non-physical locations [49].

To equilibrate at 100 K, two protocols were followed. Protocol 1 is a cooling protocol: Starting from the last frame of a 100 ns long 300 K trajectory (with TIP4P/2005 water), cooling by 10 K increments was performed. After each cooling step, a 100 ps or 1 ns NPT equilibration was performed. After 20 such steps, 100 K was reached. In protocol 2, the 168 crystal waters were retained, and converted during minimization to either the TIP3P or TIP4P/2005 water models. The Gromacs solvate command was used to add water molecules to the simulation box, while avoiding overlaps with existing molecules. NVT-NPT equilibration was performed as above, while keeping the temperature at 100 K.

For all temperatures, the production run then followed, with a 100 ns long NPT trajectory. The fastest time modes were eliminated by restricting the motion of the hydrogen atoms using the linear constraint solver (LINCS) algorithm [50]. This allowed for a relatively large time step of 2 fs, enabling us to reach a longer production time, 100 ns (0.5 × 10${}^{8}$ time steps). Such long times were needed for proper averaging over protein modes. The trajectorys’ coordinates were saved every 1 to 10 ps. These are the data passed over to the analysis phase.

Trajectory processing was performed using the visual molecular dynamics (VMD) package [51], along with our own Tcl scripts (collected in the Supplementary Documents). We determine the water structure in the hydration shells by calculating the radial distribution function (RDF) denoted by g(r), and its integral, as implemented in VMD. Alternately, we determine the average hydration numbers in both first and second solvation shells of the Res-CO oxygen atoms (where Res = Gly or Pro) from Tcl script S1.

The hydration pattern is explained, in part, from the solvent-accessible surface area (SASA) around the Res-CO oxygen, calculated from our Tcl script S2. Additionally, we analyzed the residence time distribution of water near Res-CO, using our Tcl script S3, to show that it extends to longer times than in liquid water, but much shorter than in ice. Thus, the high order of the hydration layer is not due to “ice-like” structures. Rather, the average radius of gyration, $R_{g}$ (Tcl script S6), indicative of cylinder-like symmetry, could be the origin of the highly ordered triple helix in the liquid state.

## 3. Results

### 3.1. Stability and Fluctuations

A common check on the equilibration of a protein in MD simulations is obtained by monitoring the root mean square deviation (RMSD) of its backbone atoms from their initial positions as a function of simulation time. As Appendix A shows, the RMSD of the [(PPG)${}_{10}$]${}_{3}$ peptide is essentially time-independent on the ns timescale, indicative of rapid equilibration on the simulation timescale. Comparison of the RMSD at 250 and 300 K in Appendix A shows only little temperature dependence, suggesting that the peptide is stable in spite of its short length. Indeed, for a collagen-like peptide with more variable amino acid triplets (Figure 5 in Ref. [52]) the calculated RMSD at 300 K was about 4 Å, whereas here it is only 2 Å (Appendix A). Moreover, at 100 K we find a notably lower RMSD of 0.27 Å. Thus, the homogeneous composition of the ${\left[{\left(\mathrm{PPG}\right)}_{10}\right]}_{3}$ peptide results in a very stable and ordered structure.

The root mean square fluctuations (RMSF) of the C-alpha atom in each residue is depicted in Figure 1A. It reveals large fluctuations only at the terminal residues, where the triple helix may partly unwind. This is seen particularly at a higher temperature (300 K) in Figure 1B. This may be the reason that the peptide appears not linear at 300 K, whereas it is notably more linear at 250 K, 100 K (not shown), and in the X-ray structure (below).

However, two to three residues from the chain’s end the RMSF decreases to nearly 1 Å. It then increases back to ca. 2 Å towards the middle of the chain, perhaps due to the bending mode of the triple helix. A similar behavior has been observed in Figure 6 of Ref. [52].

### 3.2. Radius of Gyration

The radius of gyration ($R_{g}$), averaged over all frames in a trajectory, is often compared with the end-to-end distance ($R_{ee}$), as both are measures for the spatial extent of a protein. Roughly speaking, $R_{g}$ is the distance from the center of the peptide to one of its ends, so it may be expected to be close to $R_{ee}/2$. However, for linear polymers generally $R_{g}<R_{ee}/2$. For example, in an “ideal chain”, $R_{g}/R_{ee}=1/\sqrt{6}=0.408$ (see Equation (12) in Ref. [53], pp. 16–17 in Ref. [54]).

Here, we calculate $R_{g}/R_{ee}$ for a linear chain such as ${\left[{\left(\mathrm{PPG}\right)}_{n}\right]}_{3}$ as a function of the number of PPG triplets, *n*. For this aim, we have simulated four additional (capped) peptides, with n=4, 5, 7, 9, and 10, each at 250 and 300 K. These numerical results are subsequently compared with a simple analytical theory that, amusingly, suggests different *n*-dependencies for even and odd *n*.

The squared radius of gyration, $R_{g}^{2}=\sum_{i=1}^{N}m_{i}r_{i}^{2}/\sum_{i=1}^{N}m_{i}$ was calculated for all *N* atoms of the peptide having masses $m_{i}$ and distances $r_{i}$ from the center of mass (CM), and averaged over all 10,000 frames of each trajectory. In polymer physics, one prefers to discuss the equal-mass case ($m_{i}=1$) [54]. Results for the five peptides at the two temperatures (and $m_{i}=1$) are collected in Table 1. Interestingly, the inclusion of mass-scaling here has a negligible effect on $R_{g}$, even more so when just the backbone is considered (see fn *a* in Table 1). The values of $R_{g}$ from experimental X-ray structures for n=9 and 10 (at about 100 K) are in excellent agreement with our simulated values, even mimicking the slight decrease in $R_{g}$ with decreasing *T* that is seen here. Given that the crystal structure is highly linear (see Figure 5 below), this agreement suggests that a linear model is indeed very suitable for the average (PPG)${}_{n}$ peptide in solution. In comparison to $R_{g}=25.5$ Å for (PPG)${}_{10}$, the *experimental* value for the (POG)${}_{10}$ peptide (using X-ray solution scattering) is only 23 Å, suggesting that a linear model is less suitable in this case [55].

Appendix A summarizes our simulated $R_{ee}$ values, measured between the two terminal α-carbons, for each chain of the ${\left[{\left(\mathrm{PPG}\right)}_{n}\right]}_{3}$ peptides (n=4, 5, 7, 9, and 10) at 250 and 300 K. Interestingly, the values of $R_{ee}$ also decrease with decreasing *T*, as observed above for $R_{g}$ (Table 1). However, these effects are small (in the third digit) and not readily interpreted.

Averaging, further, over the three chains at 300 K, yields the values in Table 2. In comparison to a ${\left[{\left(\mathrm{POG}\right)}_{9}\right]}_{3}$ simulation that gave an average $R_{ee}=72.1$ Å at 263.5 K [28], our value for ${\left[{\left(\mathrm{PPG}\right)}_{9}\right]}_{3}$ (at 250 K) is 72.7 Å. However, in comparison to both simulation results, the $R_{ee}$ from the X-ray structures (for n=9 and 10) are notably smaller (values in parentheses in Table 2), which contrasts with the high level of agreement between $R_{g}$ from X-ray and simulations in Table 1.

We suggest a simple model that explains the (PPG)${}_{n}$ results. The model collagen peptide is composed of equal mass points representing PPG triplets, arranged along a linear axis, with fixed distances *l* between them (Figure 2). Due to symmetry, it suffices to consider the *k* points on just one side of the CM.

Consider first the case of an odd number of triplets, n=2k+1, when the CM is at the central triplet of the chain. The distances of the triplets from the CM are integer multiples of *l*, namely: l,2l,...,kl, so that $R_{ee}=2kl$. Let the index *i* count these triplets, where i=0 denotes the central triplet at the CM location. Then, the averaged radius of gyration over all (equal-mass) triplets is
(1)$$R_{g}^{2}=\frac{1}{k}\;\sum_{i=1}^{k}{\left(il\right)}^{2}\;.$$
It is well known that the series of squared integers sums analytically
(2)$$P_{k}\equiv \sum_{i=1}^{k}i^{2}=\frac{1}{6}\;k\left(k+1\right)\left(2k+1\right)\;,$$
$P_{k}$ is sometimes called the “square pyramidal number” because it counts the number of identical balls that can be arranged in the form of a *k*-level square pyramid. Therefore, Equation (Equation 1) becomes:(3)$$\frac{R_{g}^{2}}{R_{ee}^{2}}=\frac{1}{6}\frac{\left(k+1)(2k+1\right)}{{\left(2k\right)}^{2}}=\frac{1}{12}\;\left(1+\frac{3}{2k}+\frac{1}{2k^{2}}\right)\;\;.$$
For example, for the ${\left[{\left(\mathrm{PPG}\right)}_{9}\right]}_{3}$ peptide k=4, and then $R_{g}^{2}/R_{ee}^{2}=0.117$, or $R_{g}/R_{ee}=0.342$.

For an even number of triplets, n=2k, $R_{ee}=\left(2k-1\right)l$, and the CM is in between the two central triplets (Figure 2). Therefore, the closest triplets to the CM are a distance l/2 from it. The averaged radius of gyration over all residues is now
(4)$$R_{g}^{2}=\frac{1}{k}\;\sum_{i=1}^{k}{\left[\left(i-1/2\right)l\right]}^{2}=\frac{1}{k}\;\sum_{i=1}^{k}\left(i^{2}-i+1/4\right)l^{2}\;,$$
obtained by replacing *i* by i−1/2 in Equation (Equation 1). Given that $\Sigma_{i=1}^{k}i=k\left(k+1\right)/2$, one obtains:(5)$$\frac{R_{g}^{2}}{R_{ee}^{2}}=\frac{1}{{\left(2k-1\right)}^{2}}\;\left[\frac{1}{6}\left(k+1\right)\left(2k+1\right)-\left(k+1\right)/2+1/4\right]=\frac{1}{12}\;\frac{2k+1}{2k-1}\;.$$
For the ${\left[{\left(\mathrm{PPG}\right)}_{10}\right]}_{3}$ peptide k=5, and then $R_{g}^{2}/R_{ee}^{2}=0.102$, or $R_{g}/R_{ee}=0.319$. This is in remarkable agreement with $R_{g}/R_{ee}=0.3145$ obtained in our simulations.

Figure 3 shows the dependence of $R_{g}/R_{ee}$ on the number of triplets in the peptide for the odd and even cases, Equations (Equation 3) and (Equation 5), respectively. Lines were added to guide the eye. The odd curve is always above the even, but at large *k* they converge to a common value, $1/\sqrt{12}=0.2887$. The triangles and squares depict the simulation results for the even and odd peptides, respectively. For n=4 and 10 the triangles fall close to theoretical points. The squares depicting n=5, 7, and 9 fall further below the odd values from Equation (Equation 3). Judging from the experimental crystal data for n=9 and 10 in Table 2, a main source of error is the disagreement with the simulated $R_{ee}$. Replacing it with the experimental $R_{ee}$ (for n=9, 10), while keeping the simulated $R_{g}$, produces the open circles Figure 3, which are in excellent agreement with the model (closed circles).

### 3.3. Characterization of Hydration Sites

In the [(Pro(X)-Pro(Y)-Gly)${}_{10}$]${}_{3}$ collagen tripeptide model chain, hydrogen bonds between the main chains are responsible for the triple-helix stability. Early fiber diffraction models already deduced the correct interstrand Gly-NH⋯OC-Pro(X) connectivity, where the NH group of the Gly residue acts as a HB donor, and the CO group of the Pro residue in the X position of an adjacent strand acts as an acceptor [14,22]. Therefore there is exactly one such HB per triplet, as depicted schematically in Figure 4.

While each bb-carbonyl group may potentially serve as a binding site for WMs (Figure 4), Pro(X)-CO cannot bind water because it already accepts a HB from a Gly-NH in a neighboring chain. Hence, only the CO of Gly and Pro(Y) are available for binding water, but they are far from equivalent. According to Bella et al. [22], Gly-CO accepts a single HB, and Pro(Y)-CO ideally accepts two. While there were disagreements concerning the [(Pro(X)-Pro(Y)-Gly)${}_{10}$]${}_{3}$ hydration structure [16], a more recent X-ray work suggests (based on PDB file 1K6F) that “the carbonyl groups of Gly residues are singly hydrated, whereas those of proline residues in the Y position are doubly hydrated” [17]. Figure 5A helps visualize the hydration structure of this tripeptide by depicting the WMs in the inner solvation shell of Gly-CO in orange and those solvating Pro(Y)-CO in yellow.

#### 3.3.1. Calculating Inner-Shell Hydration Numbers

How does one calculate the number of water neighbors in the inner solvation shell of Res-CO? We have utilized one of three routes for achieving this:(a)For a single trajectory frame, or an X-ray structure PDB file, one can simply visualize the binding site and its ligands, e.g., (i) “chain A and resname PRO and name O” in one color, and (ii) “water and within 3.5 of (chain A and resname PRO and name O)” in another color. Note that for a single frame a cutoff distance of 3.5 Å is selected, which is slightly larger than the value of 3.25 Å from an ensemble average.(b)The average number of WM neighbors of a given Res-CO site can be calculated from Tcl script S1. Here, the cutoff distance, 3.225 Å, is taken from the first minimum in g(r) for liquid water (at 300 K).(c)The equilibrium water structure at the protein surface can be captured by the radial distribution function (RDF), denoted g(r), which monitors the average number of water oxygen atoms ($O_{w}$) within a spherical shell surrounding the oxygen atom of the bb-carbonyl of each residue of the [(PPG)${}_{10}$]${}_{3}$ peptide. The Pro(X) residue did not show any peak for the first hydration layer (r<3.25 Å), because of its carbonyl oxygen HB formation with the Gly-NH of the neighboring chain. This leaves the Gly-CO and Pro(Y)-CO sites for investigation.

Figure 6A,C show the RDF of $O_{w}$ around the glycine and proline(Y) carbonyl oxygens, respectively. The positions of the first two peaks and two minima in all systems are listed in Appendix A. The probability for observing an $O_{w}$ atom in the inner hydration layer peaks at the first maximum, which is 2.65 Å in all cases (Gly/Pro(Y) at 250/300 K), extends up to the first minimum, at 3.25 Å. This value agrees also with the first minimum of the oxygen–oxygen RDF obtained from bulk water simulations 3.225 Å, as well as with experiment (Figure S4) [56].

A similar g(r) from a simulation using the popular TIP3P water model is shown in Appendix A. In comparison with the TIP4P/2005 water model used herein, it exhibits weaker undulations, particularly in the second hydration layer, which are in worse agreement with bulk water experiments, as demonstrated in Appendix A. Appendix A shows a comparison with g(r) from a simulation using the CHARMM36m FF. It, again, exhibits weaker undulations (smaller first and second peaks) in comparison with the AMBER14SB FF utilized herein. Thus, g(r) from the AMBER14SB+TIP4P/2005 combination has the most structure, and this may be affecting the hydration numbers reported below.

The indefinite radial integral of g(r), denoted n(r), gives the average number of $H_{2}O$ neighbors up to distance *r*. To obtain the hydration numbers for the inner hydration shell, we calculate n(r) at r=3.25 Å, as demonstrated in Figure 6B,D. The result is the same as from our Tcl script S1, but it requires calculating g(r) separately for each O atom in a Res-CO selection, while the Tcl script outputs the average hydration numbers for the whole selection at once.

#### 3.3.2. Crystal Structure First-Shell Hydration

The most recent X-ray structure for [(PPG)${}_{10}$]${}_{3}$ is reported as PDB file 1K6F (1.3 Å resolution) [17]. It was crystallized out of a polyethylene glycol (PEG) buffer solution near room temperature [57]. In comparison, [(PPG)${}_{9}$]${}_{3}$ (PDB file 2CUO) was crystallized from an aqueous buffer solution and flash cooled to 100 K under a vaporized nitrogen stream [19]. Therefore, aqueous MD at T=100 K should mimic the crystal structure. Hence, we calculated the inner-shell hydration numbers of the [(PPG)${}_{10}$]${}_{3}$ crystal from either one of the two 100 K simulation protocols described in the Section 2 using methods (a) or (b) in Section 3.3.1 above.

The orange circles in Figure 7A depict the number of water ligands as a function of residue number for the [(PPG)${}_{10}$]${}_{3}$ X-ray structure (PDB file 1K6F). A single WM is missing on Pro17 and Gly24 and two are missing on Pro29 (the chain end), but otherwise a perfect 0-2-1 hydration pattern emerges, as anticipated by [17,22]. A similar structure is also seen for [(PPG)${}_{9}$]${}_{3}$ in Appendix A.

The stars in Figure 7A depict similar results from our MD simulations using cooling protocol 1 (10 K cooling steps followed by 100 ps equilibration). Upon averaging over the 100 ns production run, the hydration numbers are 0-2-1 for each triplet, with the exceptions of the three residues Pro2, Pro20, and Pro29. This conforms better to the 0-2-1 regularity than the X-ray structure because only half of a WM is missing on residue 20. When the cooling steps are followed by 1 ns (instead of 100 ps) equilibration, Pro8 rather than Pro20 is deviant.

Figure 7B shows the MD hydration numbers with protocol 2: Retaining the crystal water, adding WMs with the Gromacs solvate command, equilibrating, and running for 100 ns at 100 K. The TIP4P/2005 water model shows only missing water ligands, now exclusively on Pro(Y) residues, whereas the TIP3P water model shows also extra WMs on Gly-CO.

We conclude that our MD simulation (particularly with TIP4P/2005) is very successful in reproducing the hydration pattern seen in the X-ray structure. Even the deviant residues are similar in number. Moreover, for the peptide simulated at 100 K the missing WMs are always the second WMs of a Pro(Y) residue. This shows that the *second* WM in the first Pro(Y)-CO hydration shell (i.e., the WM that binds last and/or leaves first) forms a weaker HB than either the Pro(Y)-CO⋯HOH or Gly-CO⋯HOH HBs, explaining the few deviant residues from the 0-2-1 hydration pattern.

#### 3.3.3. Room Temperature First-Shell Hydration

As the temperature was raised from 100 K and closer to room temperature (250 and 300 K), we anticipated that the high order witnessed for the first-shell WMs would disappear. Surprisingly, Figure 8 shows that at room temperature the hydration structure is similar to the crystal structure at 100 K. In both cases, the hydration numbers of Pro(X)-CO and Gly-CO are 0 and 1, whereas that of Pro(Y)-CO, which is 2 in the crystal state, is in the range 1.6–1.7 here. Moreover, the first two values are nearly temperature independent, while that of Pro(Y)-CO is slightly enhanced by lowering the temperature.

Similar results for all three chains are given in Appendix A, and against the residue number of Gly and Pro(Y) separately in Appendix A. In a similar vein, Appendix A shows that only the Pro(Y)-CO hydration at 300 K depends on the water model, becoming too small when TIP4P/2005 is replaced by TIP3P. The average over all CO sites in the protein is given in Table 3.

It can be appreciated that our hydration numbers for Pro(Y)-CO are indeed close to the crystal data in comparison with previously reported hydration numbers of ${\left(\mathrm{POG}\right)}_{n}$ peptides having their Hyp-CO coordination number around 1.3 [11,22]. Recent MD simulations of a more complex tripeptide, having the ${\left(\mathrm{POG}\right)}_{n}$ structure only at its two ends, reported an average of 1.26 waters bound to the Hyp-CO site [34]. To our knowledge, no study has previously suggested a hydration number so close to 2 for Pro(Y). We interpret this as evidence for the high order of the ${\left(\mathrm{PPG}\right)}_{n}$ hydration layer, which is in line with the high order of the tripeptide itself, as deduced from our $R_{g}$ analysis (above).

#### 3.3.4. Solvent-Accessible Surface Area (SASA)

How can one explain the remarkable regularity of water hydration at the CO sites? To further explore this, we present an analysis using the solvent-accessible surface area (SASA). It is obtained by rolling a sphere with the water radius, 1.4 Å, on the surface of the peptide [58]. The SASA measures the available space above the surface into which a WM could fit–a purely geometrical property. We calculated the SASA using the VMD “measure sasa” command, as embedded in Tcl script S2. The time averaged SASA values for each bb-O atom in the tripeptide are collected in Appendix A (300 K) and S5 (250 K).

Figure 9 shows these values for each consecutive residue of a given peptide chain. The orderly sawtooth structure testifies to an ordered peptide structure with a repetition unit of three. The SASA for the Pro(X)-CO oxygen atoms is zero, just like their coordination numbers, while the SASA for Pro(Y)-CO is slightly over twice that of Gly-CO, similar to the ratio of their coordination numbers. When the SASA values are further averaged over all like residues, one obtains the results in Table 4.

Figure 10 shows that the SASA of the carbonyl oxygen atoms is highly correlated with the number of WMs around it (Pearson correlation, $R^{2}$= 0.9582 at 300 K). Thus, the primary explanation for the 0-2-1 solvation pattern is geometric, i.e., existence of available space for water entry. However, the deviation from linearity could indicate that additional factors are operative, producing the under-coordinated Pro(Y). For example, the water–carbonyl HB strengths: A double-donor configuration, $O_{w}H\cdots O\cdots {\mathrm{HO}}_{w}$, is known to be less stable (weaker HB) than a donor–acceptor configuration, $O_{w}H\cdots O_{w}H\cdots O$, so that it is preferable for the second ligand to move to the second hydration shell.

It is instructive to compare, for each residue, the SASA of the single carbonyl oxygen atom with that of the whole residue. For whole residues, Figure 11 shows a similar sawtooth regularity, but with completely different SASA values (Table 5). Now, the SASA of Pro(X) is no longer close to zero, but rather nearly as large as for Pro(Y). This follows because the SASA of a Pro residue is contributed mainly from its side-chain ring. In contrast, the SASA of the Gly residue is only slightly larger than the Gly-CO SASA, likely because most of this residue’s atoms are close to the CMP’s axis, and thus shielded from the solvent. The structural HB between Gly-NH and Pro(X)-CO may direct some of the Pro(X) atoms to the peptide axis as well, and this may explain why its SASA is still somewhat smaller than that of Pro(Y).

#### 3.3.5. Hydrogen-Bonded Second Hydration Shell

The first hydration shell around the oxygen atom of Res-CO (Res = Gly or Pro) was established as detailed in Section 3.3.1 using a RDF, herein $g_{1}\left(r\right)$, of water around the carbonyl O atom. By convention, the first shell ends at the first minimum of $g_{1}\left(r\right)$ that occurs (as in bulk water) at $r_{\min}=3.225$ Å. The number of neighbors is then the integral of $g_{1}\left(r\right)$ up to $r_{\min}$. There are now two ways to extend the analysis to a second hydration shell:(a)Use the same $g_{1}\left(r\right)$ as above, and integrate its second peak namely, from $r_{\min}$ to the second minimum, which occurs around 5.61 Å (see Figure 6). “Second neighbors” by this criterion will include WMs that are not HBed to the water in the first hydration shell (i.e., to Res-CO⋯HOH).(b)Restricting our interest to WMs that are directly HBed to first-shell waters requires that we consider another RDF, $g_{2}\left(r\right)$, that is centered on a first-shell WM of Res-CO. The integral of $g_{2}\left(r\right)$ up to its first minimum, $r_{\min}$, is subsequently the number of HBed WMs in the second layer of Res-CO.

Using the second criterion, we obtain the (time-averaged) results shown in Figure 12. When these are further averaged over all like residues in the peptide, the values of Table 6 are obtained. Pro(X)-CO does not display any second-shell hydration characteristics because it has no first-shell hydration layer. Gly-CO has slightly over two second-shell neighbors (i.e., Gly-CO⋯HOH has slightly over two first-shell neighbors). It is possible (though we did not check this) that these are mostly two HB-donating WMs. Pro(Y)-CO has about 1.5 neighbors in the first shell. If each of these has about two neighbors in the second shell, a total of three second-shell neighbors are obtained. We also visualized that when a second WM HBs to Pro(Y)-CO, it often forms an HB “bridge" with the adjacent Gly-CO⋯HOH.

### 3.4. Residence Time Distribution of Water Near Res-CO

The results presented thus far raise a perplexing question: Why is Res-CO (and, particularly, Gly-CO) hydration so uniform (varying so little with residue number or temperature)? One possibility is thermodynamic in nature. According to Fullerton [25] (see Section 1), water bridges in collagen are so strong that they “exhibit dynamic and thermodynamic properties of one-dimensional ice” [34]. In particular, this should apply to the strong Res-CO⋯HOH HBs. Indeed, we find that the 0-1.7-1 hydration pattern of [(PPG)${}_{10}$]${}_{3}$ in liquid water is similar to the 0-2-1 pattern in ice. However, to explain Fullerton’s NMR data, the “one-dimensional ice” theory should be valid at least up to the msec time-regime, while no MD simulation reported HB lifetimes longer than, say, 10 ps [31,33,34].

The second possibility is kinetic: While water dissociation from the Res-CO site is fast, the recombination by an alternate WM is even faster, so that the binding site is rarely ever vacant. Here, we set out to investigate these possibilities.

The residence time, τ, measures the time that elapsed between a WM arriving and leaving a binding site, such as Res-CO, where a water at the site is defined by the CO⋯O${}_{w}$ distance being less than a cutoff distance (3.5 Å). We do not measure the distance to the water H atoms because these can interchange without the water oxygen, O${}_{w}$, leaving its bound position (water libration). Moreover, if a water ligand leaves the site for a single time step, returning in the next step, the binding episode is not discontinued. Only moving outside the cutoff distance for more than a single time step ends the sojourn. These are the underlying assumptions in the Tcl script S3, utilized herein.

The list of τ values obtained this way spans a wide range of times. Averaging over the list gives the average residence time, <τ>. Otherwise, the τ values are binned, the number of occurrences in each bin depicting a temporal correlation function, f(t), as shown in Figure 13. Fitting it to an exponential function, f(t)=Aexp(−t/<τ>), is another way of obtaining <τ>. This follows because the exponential distribution obeys $<\tau >=\int_{0}^{\infty }tf\left(t\right)dt/\int_{0}^{\infty }f\left(t\right)dt$.

The fitting parameters, *A* and <τ>, are listed in Table 7. Overall, water at Gly-CO has longer residence times than at Pro(Y)-CO, which is in line with a stronger Gly-CO⋯HOH HB. In Pro(Y)-CO, it is mostly the second water ligand that leaves, whose HB is weaker than the first one. For the same reason (see above), the coordination number of Gly-CO varies less with residue number or temperature as compared to Pro(Y)-CO. We have noted that a WM departure from the Res-CO site was always accompanied by the binding of another WM, so essentially there is no free site in this scenario, which is best characterized as a substitution reaction.

Returning to the question of “ice-like” HBs, this study agrees with previous MD studies, in which no ice-like HBs were identified. Even the strongest water-collagen HB, Gly-CO⋯HOH, exhibits a sub-100 ps lifetime (much faster than the NMR timescale [59]). On the other hand, this lifetime is notably longer than the average lifetime of 4 ps previously calculated [34], longer also than either the single-molecule or cooperative Debye relaxation times in bulk liquid water (ca. 1 ps and 8.3 ps at 300 K, respectively [60]).

## 4. Conclusions

Collagen, with its peculiar triple-helix structure, is one of the major proteins in the ECM. It is imperative to achieve a better understanding of the dynamic consequences of its structure, not only for coping with living organisms, but possibly also for seeking inspiration concerning the construction of man-tailored materials. Unfortunately, it is not possible to isolate even a few of the many collagen structures for laboratory study. Much of what we currently know, on the molecular level, is based on synthetic collagen biomimetic peptides. Such is the [(PPG)${}_{10}{]}_{3}$ 90-residue peptide (30 in each chain), which is the prime model system for the current theoretical study.

Single-chain proteins can partially fold to assume a nearly spherical (globular) shape, in which case $R_{g}$ is the radius (in Å) of the globule. It increases rather slowly with the number of amino acid residues, *N*, as [61]
(6)$$R_{g}=0.395\;N^{3/5}+7.257$$
A similar power law correlation has also been established with the proteins’ molecular weight [62]. In contrast to globular proteins, an elongated protein will have a notably larger $R_{g}$. For example, setting N=90 in Equation (Equation 6) gives 13.1 Å, in comparison to $R_{g}=25.6$ Å obtained here for [(PPG)${}_{10}$]${}_{3}$ (see Table 1). Thus, the (PPG)${}_{10}$ tripeptide is clearly an elongated molecule rather than a spherical one.

This, however, does not capture its high symmetry. To establish it quantitatively, we have presented, in Section 3.2, a simple model for a linear repetitive molecule. The *n*-unit tropocollagen is depicted as a rigid linear array of equidistant (distance *l*) points of equal mass (PPG triplets). This results in two analytical expressions for $R_{g}$ as a function of *n*, to be used for odd and even *n*:(7)$$R_{g}^{2}=\frac{l^{2}}{12}\;\{\begin{array}{ccc} \; & (n+1)n\; & \mathrm{odd},\\ \; & (n+1)(n-1)\;\;\; & \mathrm{even}, \end{array}$$
The distance *l* between beads can be obtained from the difference in $R_{ee}$ for simulated peptides of different lengths (Table 2). For example, from $R_{ee}$ for n=10 and 9 l=81.41−72.56=8.85 Å (at 300 K). With this, $R_{g}\left(n\right)$ is given in Figure 14 which, in contrast to the power-law in Equation (Equation 6), is practically linear.

From Equation (Equation 7) we find that for n=9 and 10 $R_{g}=24.24$ and 25.42 Å, respectively, in excellent agreement with our simulated values (24.0 and 25.6 Å) in Table 1. Thus, the depiction of the (PPG)${}_{n}$ peptide as a linear array of PPG triplets is in good agreement with the simulations.

A central goal of the present work is to characterize the hydration pattern in a “simple” collagen tripeptide, [(PPG)${}_{10}$]${}_{3}$. The careful $R_{g}$ analysis performed in Section 3.2 suggests that the tripeptide resembles a cylinder on which the amino-acids are arranged. Because this cylinder is not appreciably bent, all Pro(X) residues are located in equivalent environments, and similarly for Pro(Y) and Gly. Given that the only water binding sites on this CMP are the bb-carbonyls, the expected outcome could be a regular hydration pattern.

Indeed, by monitoring the number of WMs bound to each Res-CO site, we have found a remarkably regular pattern. Pro(X)-CO does not bind any WM, and Gly-CO binds approximately one. In the X-ray structures, as well as our MD simulations at 100 K, Pro(Y)-CO binds nearly always 2 WMs. At room temperature this reduces to 1.6–1.7 WMs. This 0-1.7-1 pattern holds for all 90 residues (30 triplets) of the triple helix (excepting those at the very ends of the chains). This behavior is close to the 0-2-1 hydration pattern observed in X-ray structures and our MD simulations at 100 K. Thus there is persistent order extending from crystals (near 100 K) to liquid water at room temperature.

To better understand this behavior, we have monitored the SASA of each oxygen atom of the tripeptide. It exhibits a pattern nearly proportional to the bb-carbonyl water coordination, of ca. 0-6.5-15 Å${}^{2}$. Thus, near the Pro(X)-CO site, there is simply not sufficient space to insert a WM. The ratio of the Pro(Y) to the Gly oxygen atom SASA values of about 2.3 shows that (as in the crystal state) there is sufficient space for two WMs near Pro(Y)-CO. The reason for the reduced coordination at the Y position must then be energetic: The second WM is bound less strongly in the Pro(Y)-CO site. This is also manifested by the enhanced fluctuations in the Pro(Y)-CO coordination number with residue index (e.g., Figure 8), as compared with Gly-CO. It follows that (at least) the first WM bound to each Res-CO site (excepting Pro(X)-CO) exhibits high spatial order.

Dynamically, there is a large difference between the peptide’s hydration in ice vs. liquid water. While the WMs seen in the X-ray structures likely exchange very slowly, this occurs at the 100 ps timescale in liquid water, as evidenced by our residence time calculations. While this is slower than suggested from previous simulations in the literature (ca. by a factor 20) [34], it is still orders of magnitude faster than in ice.

How can the bound WMs show such regularity in their coordination numbers in spite of their fast dynamics? We find that a bound WM leaving the site is immediately replaced by another. Thus WMs undergo an exchange, i.e., substitution, rather than a dissociation reaction. Consequently, there is always a WM at the site, resulting in highly ordered Res-CO hydration in spite of the high temperature.

In contrast to the strong HBs found between WMs and the bb-carbonyls, we have not discovered in room temperature [(PPG)${}_{n}$]${}_{3}$ peptides inter-helix water bridges, or ice-like water bridges, which are only detectable via X-ray [25]. Thus, at least from the simple CMP studied here, we cannot claim that water wires contribute to the stability of collagen. Conversely, the rigidity of collagen contributes towards an exceptional ordering of its (mainly, first-shell) hydration water. Future work would seek to determine possible consequences of the high hydration-layer order for collagen dynamics.

## Figures and Tables

**Figure 1 biomolecules-13-01744-f001:**
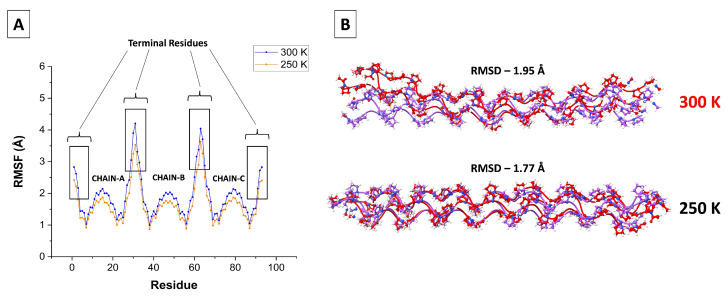
(**A**) The root mean square fluctuation (RMSF) for each α carbon of the [(PPG)${}_{10}$)]${}_{3}$ peptide at 250 K (orange) and 300 K (blue), with the data for the three chains presented successively. (**B**) Comparison of superimposed triple-helix structures at different temperatures: Initial configuration in purple and final structure in red. AMBER14SB+TIP4P/2005, 62,306 WMs, 100 ns trajectory (10,000 saved frames).

**Figure 2 biomolecules-13-01744-f002:**
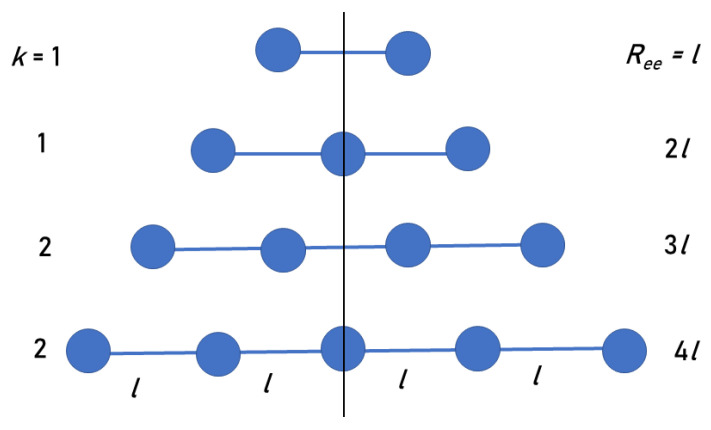
The first four smallest linear peptides in our model (two even and two odd), whose CMs are along the vertical line. The values of *k* and $R_{ee}$ for each peptide chain are indicated in the figure. Each value of *k* appears twice: For even and odd *n*, giving rise to the even and odd series in our model.

**Figure 3 biomolecules-13-01744-f003:**
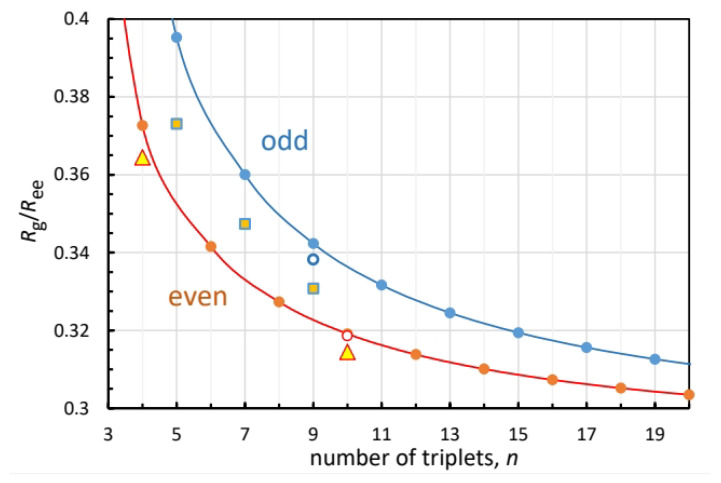
Ratio of radius of gyration to end-to-end distance as a function of the number of PPG triplets (circles), calculated from Equations (Equation 3) (blue) and (Equation 5) (brown) of our analytical model. Lines represent interpolations. Simulated values for odd and even [(PPG)${}_{n}$]${}_{3}$ peptides are depicted by squares and triangles, respectively. Open circles are ratios of simulated $R_{g}$ to the X-ray $R_{ee}$ values.

**Figure 4 biomolecules-13-01744-f004:**
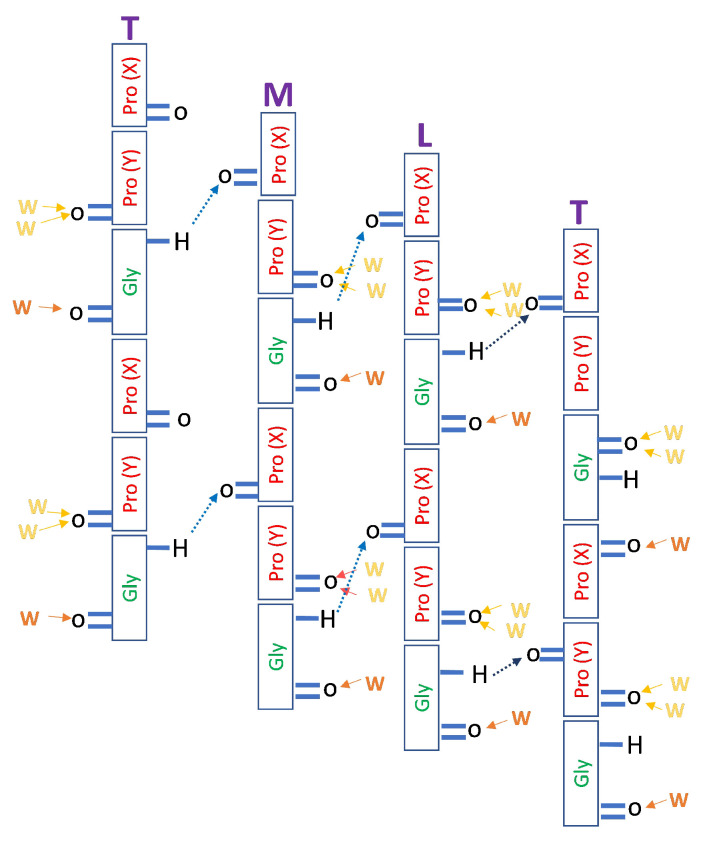
Schematic HBs of the [(PPG)${}_{10}$]${}_{3}$ peptide. The three chains forming the triple helix are conventionally denoted as trailing (T), middle (M), and leading (L), with the T chain repeated on the right to show the interactions more clearly. Dashed blue arrows depict the structural Gly-NH⋯OC-Pro(X) HBs. The WMs shown resemble those found in the X-ray diffraction studies: The single orange WM binds to Gly-CO, while the two yellow ones bind to Pro(Y)-CO.

**Figure 5 biomolecules-13-01744-f005:**
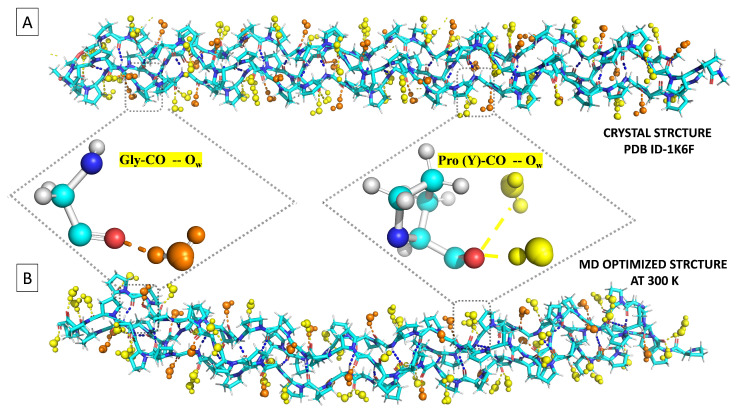
Comparison of (**A**) the X-ray structure of the [(PPG)${}_{10}$]${}_{3}$ peptide (PDB 1K6F) with (**B**) a snapshot from the end of a 100 ns MD simulation at 300 K. Prepared with the VMD package [51].

**Figure 6 biomolecules-13-01744-f006:**
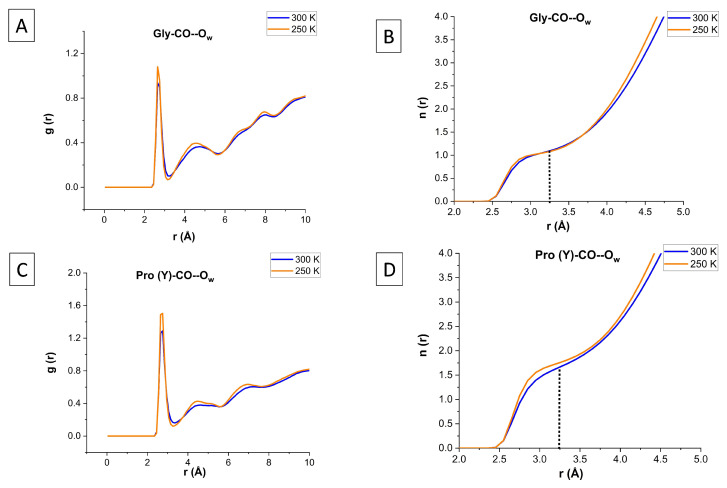
RDF profiles (**A**,**C**) and their indefinite integrals (**B**,**D**) for water surrounding the carbonyl oxygens of the Gly and Pro(Y) residues of [(PPG)${}_{10}$]${}_{3}$ at 300 K (blue) and 250 K (transparent orange). AMBER14SB+TIP4P/2005, 62,306 WMs, 100 ns simulation (10,000 frames).

**Figure 7 biomolecules-13-01744-f007:**
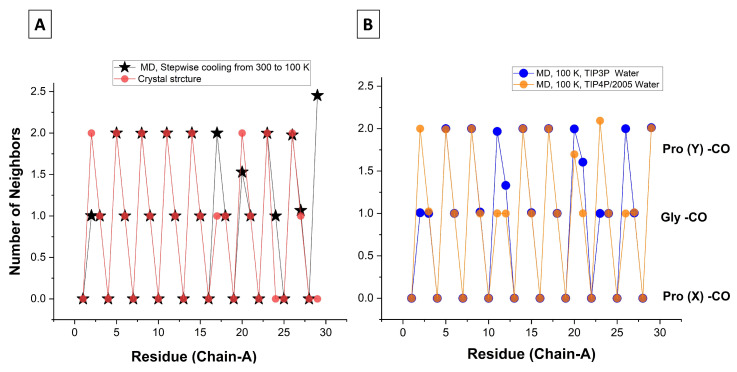
Number of WMs observed in the bb-CO sites of the [(PPG)${}_{10}$]${}_{3}$ as a function of residue number (Pro(X) zero, Gly one, and Pro(Y) two). (**A**) Transparent orange circles: X-ray data from the α chain in PDB file 1K6F [17] (1.30 Å resolution). Black stars: MD with the TIP4P/2005 water model, cooling to 100 K according to protocol 1 (see Section 2), and averaging over the ensuing 100 ns production run. (**B**) Results from protocol 2. Blue circles in background: TIP3P water; Transparent orange circles in foreground: TIP4P/2005 water model. Note: When both circles coalesce, the color appears dark orange.

**Figure 8 biomolecules-13-01744-f008:**
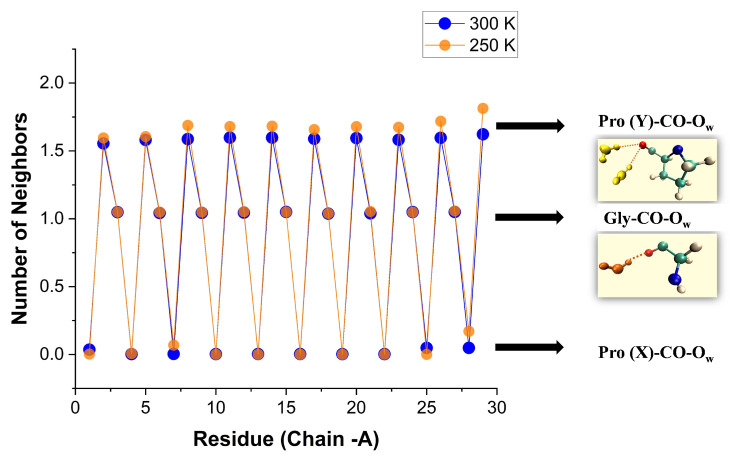
Number of water neighbors in the first solvation shell of the Res-CO sites on the α-chain of the [(PPG)${}_{10}$]${}_{3}$ peptide, obtained from integrating g(r) at two different temperatures, 250 K (transparent orange) and 300 K (blue) (AMBER14SB+TIP4P/2005, 62,306 WMs, 100 ns simulation (10,000 frames)). Results for all three chains appear in Appendix A.

**Figure 9 biomolecules-13-01744-f009:**
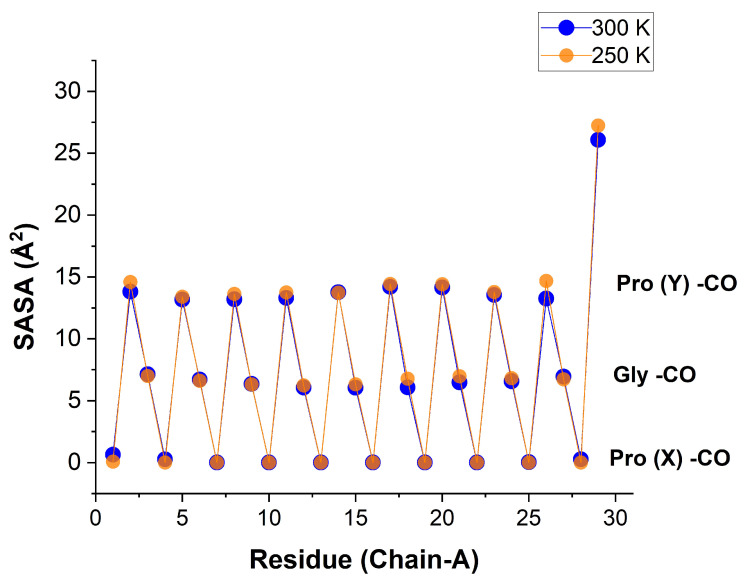
Solvent-accessible surface area (in Å${}^{2}$) for the bb-carbonyl oxygen atoms of chain α residues of the [(PPG)${}_{10}$]${}_{3}$ peptide at 250 K (transparent orange) and 300 K (blue). Data for all three chains are listed in Appendix A, and plotted in Appendix A. Separate plots for SASA values for Gly and Pro(Y), against the residue numbers are shown in Appendix A. A comparison of the results for the TIP3P vs. TIP4P/2005 water models is presented in Appendix A (AMBER14SB+TIP4P/2005, 62,306 WMs, 100 ns trajectory (10,000 frames)).

**Figure 10 biomolecules-13-01744-f010:**
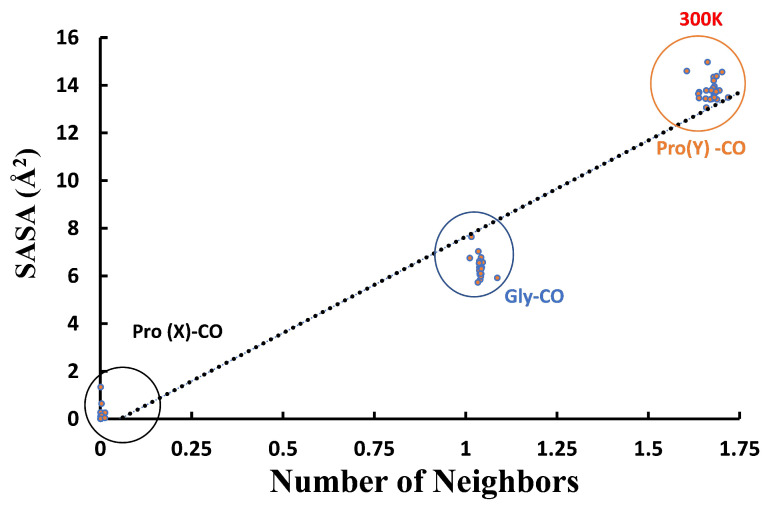
Correlation plot of the number of water neighbors of the Res-CO oxygen atom, with its SASA value in our simulations of the [(PPG)${}_{10}$]${}_{3}$ peptide triple helix at 300 K (AMBER14SB+TIP4P/2005, 62,306 WMs, 100 ns trajectory (10,000 frames)).

**Figure 11 biomolecules-13-01744-f011:**
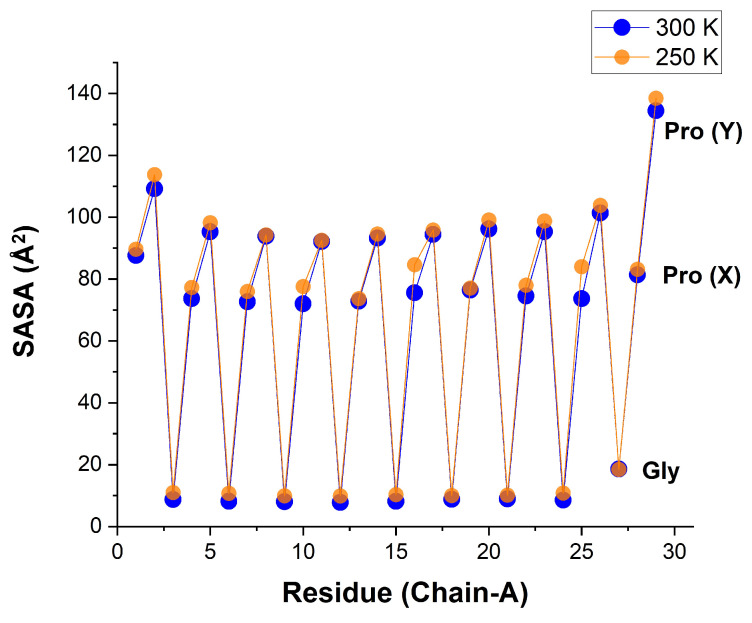
Solvent-accessible surface area (in Å${}^{2}$) for each residue in the α chain of the hydrated [(PPG)${}_{10}$]${}_{3}$ peptide at 250 K (transparent orange) and 300 K (blue) (AMBER14SB+TIP4P/2005, 62,306 WMs, 100 ns trajectory (10,000 frames)).

**Figure 12 biomolecules-13-01744-f012:**
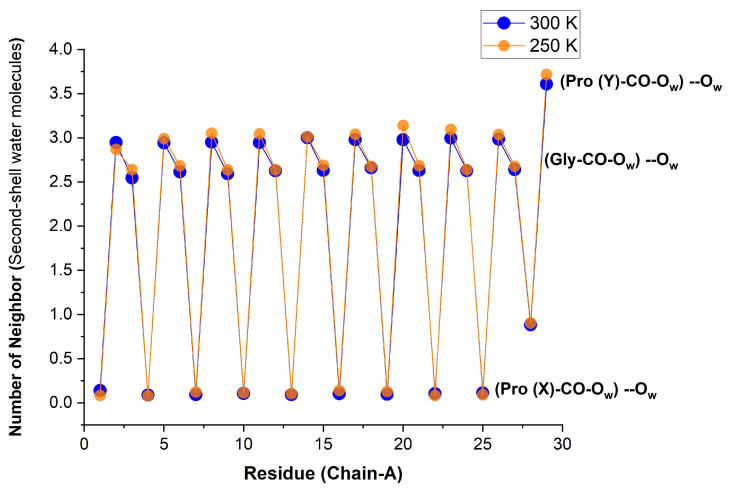
Time-averaged number of WMs that are HBed to the inner-shell water ligands of Res-CO for chain α residues of the [(PPG)${}_{10}$]${}_{3}$ peptide at two different temperatures. Obtained by integrating the RDF $g_{2}\left(r\right)$ in VMD up to $r_{\min}$, where $g_{2}\left(r\right)$ is defined by selection 1 = “atom $O_{w}$ and within $r_{\min}$ of residue Res and name O”, selection 2 = “atom $O_{w}$”. AMBER14SB+TIP4P/2005, Water-62,306, 100 ns (10,000 frames).

**Figure 13 biomolecules-13-01744-f013:**
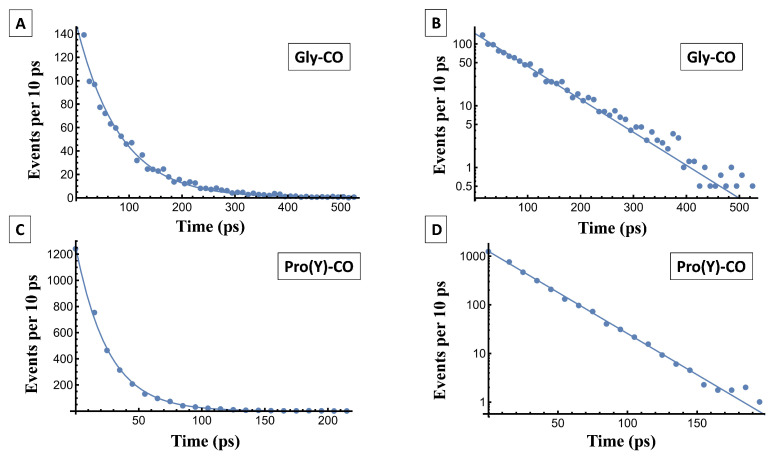
The time-dependent residence probability of WMs near the Res-CO oxygen atom, (**A**) Gly-CO and (**C**) Pro(Y)-CO at 300 K. Circles represent the simulation data, while the solid line is a mono-exponential fit to it. (**B**,**D**) show these data on a semi-logarithmic scale, in which the straight lines reveal the underlying exponential decay. The parameters of the fit are listed in Table 7.

**Figure 14 biomolecules-13-01744-f014:**
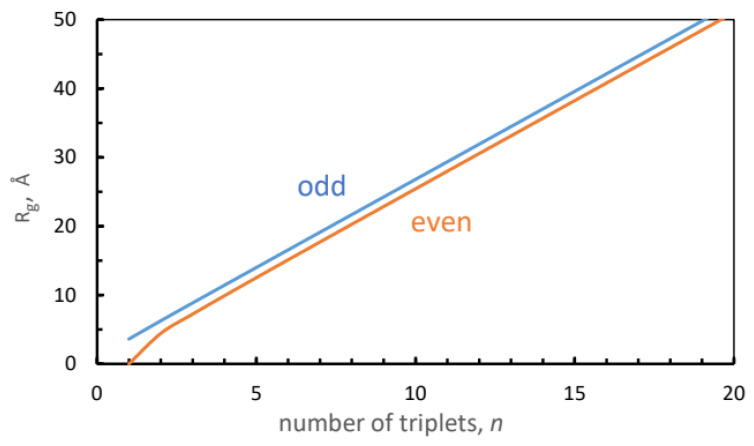
Radius of gyration calculated from Equation (Equation 7). See Section 3.2.

**Table 1 biomolecules-13-01744-t001:** Time-averaged radius of gyration, $R_{g}$ (in Å, without mass scaling), for five collagen mimetic peptides, ${\left[{\left(\mathrm{PPG}\right)}_{n}\right]}_{3}$, n=4, 5, 7, 9, and 10, simulated in aqueous solution at two different temperatures (300 K and 250 K), and further averaged over the three chains. For n=9 and 10, comparison is made with X-ray data from PDB files 2CUO and 1K6F, respectively.

Temperature	n=4	n=5	n=7	n=9	n=10
300 K	11.95	14.49	18.9	24.0	25.6 ${}^{a}$
250 K	11.91	14.36	18.96	23.9	25.5
100 K (X-ray)				23.51	25.37

${}^{a}$ With mass scaling $R_{g}=25.51$ Å in VMD and 25.57 Å in Gromacs. When only the backbone is considered, $R_{g}=25.1$ Å in VMD both with and without mass scaling.

**Table 2 biomolecules-13-01744-t002:** Averaged end-to-end distances, $R_{ee}$ (in Å), calculated from our MD trajectories for five collagen-like peptides, ${\left[{\left(\mathrm{PPG}\right)}_{n}\right]}_{3}$, in aqueous solution at 300 K, and the corresponding values of $R_{ee}/R_{g}$. $R_{ee}$ values are from Appendix A, further averaged over the three chains. In parentheses are the n=9 and 10 entries calculated using the X-ray $R_{ee}$ values at ca. 100 K from PDB files 2CUO and 1K6F, respectively (in place of the simulated values).

	n=4	n=5	n=7	n=9	n=10
$R_{ee}$	32.78	38.84	54.40	72.56 (70.66)	81.41 (80.03)
$R_{g}/R_{ee}$	0.3645	0.3731	0.347	0.3308 (0.3382)	0.3145 (0.3186)

**Table 3 biomolecules-13-01744-t003:** Average coordination numbers in the first hydration shells of Gly-CO and Pro(Y)-CO at two different temperatures (AMBER14SB+TIP4P/2005, 62,306 WMs, 100 ns trajectory (10,000 frames)).

Temperature	Water/Gly-CO	Water/Pro(Y)-CO
300 K	1.041	1.61
250 K	1.056	1.70

**Table 4 biomolecules-13-01744-t004:** Time and ensemble average of the solvent-accessible surface area (SASA) for the oxygen atoms of the carbonyl groups of respective residues at two temperatures (AMBER14SB+TIP4P/2005, 62,306 WMs, 100 ns trajectory (10,000 frames)).

Temperature	Gly-CO (Å${}^{2}$)	Pro(Y)-CO (Å${}^{2}$)
300 K	6.43	15.03
250 K	6.56	15.85

**Table 5 biomolecules-13-01744-t005:** Time and ensemble average of the solvent-accessible surface area (SASA) for each residue in the α chain of the hydrated [(PPG)${}_{10}$]${}_{3}$ peptide at 250 and 300 K (AMBER14SB+TIP4P/2005, 62,306 WMs, 100 ns trajectory (10,000 frames)).

Temperature	Gly (Å${}^{2}$)	Pro(X) (Å${}^{2}$)	Pro(Y) (Å${}^{2}$)
300 K	8.45	74.71	96.09
250 K	10.57	78.79	101.87

**Table 6 biomolecules-13-01744-t006:** Average hydration numbers in the second shell of Res-CO, restricted to WMs that are HBed to the first coordination shell ${}^{a}$ (AMBER14SB+TIP4P/2005, Water-62,306, 100 ns (10,000 frames)).

Temperature	Water/Gly-CO	Water/Pro(Y)-CO
300 K	2.26	2.93
250 K	2.37	3.01

${}^{a}$ Method (b). In comparison, method (a) for the sum over the two solvation shells (integrating $g_{1}\left(r\right)$ up to 5.61 Å) gives 6.97 neighbors for Gly and 7.96 neighbors for Pro(Y). Even Pro(X) has 2.59 non-HB neighbors.

**Table 7 biomolecules-13-01744-t007:** The fitting parameters of the residence time distribution for water to reside up to 3.5 Å from a carbonyl oxygen atom. The simulation data are fitted using a mono-exponential function, f(t)=Aexp(−t/<τ>), where *A* is the pre-exponential constant and <τ> is the time constant.

System	*A*	<τ> (ps)	$R^{2}$
Gly-CO (300 K)	129.23	81.72	0.98
Gly-CO (250 K)	5.88	129.17	0.69
Pro(Y)-CO (300 K)	1258.0	25.68	0.99
Pro(Y)-CO (250 K)	590.81	39.34	0.98

## Data Availability

Data are generated by authors following conventional MD protocols. Wherever analysis required in-house Tcl scripting, these scripts are reproduced in the Supporting Information.

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
