# Peer review of "Collagen Structured Hydration"

_biomolecules, 2023, doi:10.3390/biom13121744_

Round 1

Reviewer 1 Report

Comments and Suggestions for Authors

The study by Satyaranjan Biswal et al reports on the water structure around collagen mimetics. The study is in principle interesting in terms of insight, however there exist issues that need to be addressed prior to the consideration for publication. In detail:
(1) In Fig. 6 the authors compare the experimental crystal structure to just one snapshot (frame) of their MD trajectories. How was this one frame chosen? An ensemble average structure would be more appropriate.

(2) In Fig. 13, panels A and D seem to have the semi-logarithmic scale, instead of the reporting B-D panels. It would be also better to have all similar scale plots in one graph for comparison,

(3) More importantly, there is a recent study-method on probing the water dynamics from MD simulations, which can be directly compared with the experimental diffraction data from the crystallography (J. Am. Chem. Soc. 2023, 145, 27, 14621-14635). The authors should employ a similar computational MD methodology for the comparison of resident waters within the experimental crystal structures in the literature. This will certainly correlate the positions of waters between MD and crystal data and further validate the conclusions.
(4) In the conclusions, the authors propose that “The ordering of the WM dipole moments around the collagen axis (with cylindrical symmetry) could generate a cylindrical electrostatic field in the water phase surrounding the triple helix. If this field is sufficiently strong, it could guide the diffusion of ions (and protons) parallel to its axis, constituting a one dimensional ion conductor”. This is not supported by the data and it is an extrapolation. If the authors want to calculate electric field (with gmx potential), or the water dipole moments orientation, to prove their case, they should consider the MD methodology in Proc. Natl. Acad. Sci. USA 2016, 113(30) 8424-8429

Based on the above, I recommend a major revision of the manuscript.

Comments on the Quality of English Language

English language quality: fine.

Reviewer 2 Report

Comments and Suggestions for Authors

Comment 1:

Agmon, N., 1995. The grotthuss mechanism. Chemical Physics Letters244(5-6), pp.456-462.

above mentioned reference should be added here after:

"As far as we currently know, water is essential for life, particularly in its liquid 15 state that allows dynamics to take place [1]."

Comment 2:

When results are available at 100 K the why not simulations are being carried out at this temperature.

Comment 3:

How differently temperature effects the radius of gyration and end-to-end radius please explain qualitatively and quantitatively as well. 

Comment 4:

I ns equilibration was first performed in NVT ensemble and then in NPT ensemble. Why this particular order (1:NVT ; 2:NPT)  was followed can it be reversed (1:NPT ; 2:NVT) if yes then comment please.

Comment 5:

In Fig 6. Please mention at which time this snapshot is taken from the MD trajectory.

Comment 6:

add some new literature for MD simulations in manuscript. following one:

Riedlová K, Saija MC, OlżyÅ„ska A, Vazdar K, Daull P, Garrigue JS, Cwiklik L. Latanoprost incorporates in the Tear Film Lipid Layer: an experimental and computational model study. International Journal of Pharmaceutics. 2023 Sep 4:123367.

Author Response

See attached files

Round 2

Reviewer 1 Report

Comments and Suggestions for Authors

The authors made no changes to the manuscript/ study, regarding my comments. At this stage I cannot recommend publication.

Comments on the Quality of English Language

No major issues identified.

Author Response

See attached PDF, red texts
